# Current Understanding of an Emerging Role of HLA-DRB1 Gene in Rheumatoid Arthritis–From Research to Clinical Practice

**DOI:** 10.3390/cells9051127

**Published:** 2020-05-02

**Authors:** Tomasz Wysocki, Marzena Olesińska, Agnieszka Paradowska-Gorycka

**Affiliations:** 1Department of Systemic Connective Tissue Diseases, National Institute of Geriatrics, Rheumatology and Rehabilitation, Spartańska 1, 02-637 Warsaw, Poland; klinika.chorobtkanki@spartanska.pl or; 2Department of Molecular Biology, National Institute of Geriatrics, Rheumatology and Rehabilitation, Spartańska 1, 02-637 Warsaw, Poland; zaklad.biochemii@spartanska.pl or

**Keywords:** rheumatoid arthritis, HLA-DRB1, polymorphisms

## Abstract

Rheumatoid arthritis (RA) is an autoimmune disease with an unclear pathogenic mechanism. However, it has been proven that the key underlying risk factor is a genetic predisposition. Association studies of the HLA-DRB1 gene clearly indicate its importance in RA morbidity. This review presents the current state of knowledge on the impact of HLA-DRB1 gene, functioning both as a component of the patient’s genome and as an environmental risk factor. The impact of known HLA-DRB1 risk variants on the specific structure of the polymorphic HLA-DR molecule, and epitope binding affinity, is presented. The issues of the potential influence of HLA-DRB1 on the occurrence of non-articular disease manifestations and response to treatment are also discussed. A deeper understanding of the role of the HLA-DRB1 gene is essential to explore the complex nature of RA, which is a result of multiple contributing factors, including genetic, epigenetic and environmental factors. It also creates new opportunities to develop modern and personalized forms of therapy.

## 1. Introduction

Rheumatoid arthritis (RA) is a chronic autoimmune disease which leads to a progressive joint destruction and disability. RA affects approximately 1–2% of the adult population, which makes it one of the most common rheumatic diseases [1]. The mechanism of RA development has not been fully uncovered yet, however, the role of genetic factors that predispose an individual to the disease after the activation of factors inducing chronic inflammation, e.g., environmental factors, appear to play a critical role.

A properly functioning human immune system requires a balance between the identification and subsequent elimination of foreign environmental antigens and maintenance of the immunological tolerance in relation to its own antigens [2]. Maintaining immunological tolerance takes place at two levels:central, in which the self-reactive B and T cells are deleted during maturation in the thymus and bone marrow, respectively,peripheral, occurring through one of three mechanisms: clonal deletion (usually via apoptosis), induction of anergy (functional inactivation without cell death), or suppression of lymphocytes activation either by regulatory T cells or by clonal ignorance [3].

A loss of tolerance occurs when autoreactive lymphocytes are not inhibited due to the above mechanisms maintaining the homeostasis. It is a result of a complex process, in which environmental factors affect genetically susceptible individuals, which may lead to the development of systemic autoimmune diseases, including RA. The autoimmune response in RA is presumably initiated by citrullination of self-peptides, leading to alterations of their properties. This leads to the activation of complex immune responses and specific anti-citrullinated protein antibodies (ACPA) generation, found in approximately 75% of RA patients [4].

The risk of developing RA largely depends on hereditary factors, i.e., the disease is polygenic [5]. Multiple genome-wide association studies (GWAS) have already uncovered over 100 genetic loci associated with an increased risk of RA [6]. The new challenge in the post-GWAS era is to unravel the roles of susceptibility loci in disease development by estimation of its contribution to the overall heritability. The estimated heritability assessed in twin studies is not conclusive and ranges from 12% to 65% [7,8,9]. Interestingly, twin studies have found no difference in heritability in subsets of ACPA-positive and ACPA-negative RA [10], which is contrary to the results of another familial aggregation study. An analysis of large Swedish population registers showed estimated heritability, accounting for around 50% for ACPA-positive RA, but only 20% for ACPA-negative RA [11].

In terms of relevance in RA pathogenesis, the most recognized part of the human genome is the human leukocyte antigen (HLA) region located on chromosome 6 (Figure 1). The region consists of genes encoding molecules responsible for regulating immune response. HLA molecules are cell surface–bound glycoproteins classified into three classes. HLA class I and III are involved in presentation peptides from inside the cell and complement activation, respectively. HLA class II is expressed on the surface of antigen-presenting cells (including macrophages, B cells and dendritic cells) and is essential in order to display peptides to T-helper CD4+ cells, inducing their activation. HLA class II antigens are encoded by DR, DQ and DP (classical) and DM, DO (nonclassical) loci. Individual amino acid position variance in HLA class II molecules, especially within HLA-DR molecules forming antigen-binding grooves, explain to a large extent the overall importance of the HLA region in RA.

Previously, the contribution of HLA locus has been estimated to be 37–50% [10,12]. Recently, novel statistical methods have been introduced for fine-mapping trait-associated genomic regions with the use of statistics from GWAS studies. A large-scale HLA fine-mapping analysis of RA in the Japanese population was performed, showing that HLA genes account for 9.2% of the phenotypic variance of ACPA-positive RA, and only 1.5% of ACPA-negative RA. Within the HLA region, the contribution of the HLA-DRB1 gene was far stronger than other HLA loci (6.4% vs. 2.8%) [13].

Until now, HLA-DRB1 genotyping has neither been used in everyday clinical practice, nor included in the current ACR/EULAR 2010 rheumatoid arthritis classification criteria. However, it has been shown that some HLA-DRB1 variants may predict the unfavorable course of the disease, including a higher risk of radiographic damage progression, higher incidence of interstitial lung disease or lymphoproliferative diseases. It also seems possible that the identification of high-risk patients with the HLA-DRB1 risk allele may be important in personalizing therapy. It was observed that early and aggressive immunosuppressive treatment brings particular benefits in patients with HLA-DRB1 risk alleles [14,15]. Further research concerning the contribution of HLA-DRB1 to the RA pathogenesis may open new pathways for enhanced diagnostics and therapy of this common, disabling disease.

## 2. The Pathogenic Link between ACPA and HLA in Rheumatoid Arthritis

Citrullination is a posttranslational modification of proteins catalyzed by peptidyl-arginine-deiminases (PADs), a calcium-dependent, intracellular group of enzymes. It results in a change of positively charged arginine to a polar—but neutral—citrulline, introducing novel epitopes on self-proteins. ACPA, generated by citrulline-specific B cells, may react with various citrullinated autoantigens, including fibrin, fibrinogen, vimentin, type II collagen, α-enolase, histones, immunoglobulin binding protein-BIP, tenascin-C. In clinical practice, ACPA detection relies mostly on the commercially available measure of antibodies against cyclic citrullinated peptide (CCP) fragments of natural human proteins. The appearance of ACPA may precede many years of the development of RA symptoms. In addition, prevalence of ACPA gradually increases up to the diagnosis [16,17].

The pathogenetic significance of ACPA in RA is a result of their multidirectional biological activities (Figure 2). Complexes consisting of citrullinated fibrinogen and ACPA, found in nearly two thirds of ACPA-positive patients, have been shown to stimulate Fcγ receptors on macrophages [18], thereby inducing the release of tumor necrosis factor-α (TNF-α), a multifunctional proinflammatory cytokine [19]. The TNF-α secretion may be further amplificated by the incorporation of IgM RF IgM rheumatoid factor (RF) into ACPA-immune complex (ACPA-IC) [20]. In addition, IgM and IgA RF fractions show potential to propagate ACPA-IC-mediated complement activation [21]. Other postulated biological effects of ACPA are: activation of complement via both the classical and alternative pathways, induction of neutrophil extracellular traps (NETs) and activation of osteoclasts [22,23,24].

Currently, it is thought that the production of ACPA is mainly determined by the presence of specific environmental factors, rather than genetic factors. This hypothesis is evidenced by the existence of an association between the presence of the HLA-DRB1 risk alleles (specifically shared epitope alleles, further described) and ACPA in people with RA, and the absence of such an association in healthy, but ACPA-positive people. Consequently, the association of HLA-DRB1 alleles and the increased risk of developing RA is may not be due to the effect on ACPA production, but rather due to the effect on the pathogenicity of these antibodies. Intriguingly, the ACPA response tends to evolve prior to the onset of RA. The number of recognized citrullinated peptides increases (which may be related to epitope spreading), and avidity maturation also takes place [26,27].

The analysis of the distinct molecular structure of ACPA is crucial to understand its contribution to RA pathophysiology. ACPA, compared to other antibodies, is characterized by contained glycans in the Fc region and N-linked glycans in their variable domains (Fab) [28,29]. The formation of N-glycosylation sites is likely to be a cause of the pathogenicity of ACPA, or it may affect their persistence. Long-term observations of RA patients indicate that the ACPA glycosylation process can be detected already more than 15 years before the RA onset and intensifies as the first symptoms of the disease approach [30,31]. The incorporation of N-glycosylation sites is thought to be a result of CD4+ T cell dependent somatic hypermutation of ACPA [32].

According to the results of recent studies, the occurrence of HLA-DRB1 risk alleles predispose the formation of N-glycosylation sites in ACPA-IgG, but the precise mechanism of this predisposition is not known [30]. Nevertheless, due to the fact that HLA-DR (encoded by HLA-DRB1 gene) molecules are necessary for the activation of CD4+ T cell, we assume that it is likely that HLA-DRB1 risk alleles induce somatic hypermutation of ACPA, leading to their pathogenicity.

## 3. Role of Hypervariable Regions

The association between HLA-DR and RA susceptibility was first reported over 40 years ago [33]. It soon became apparent that the HLA-DR complex is highly polymorphic, especially in the DRβ1 chain encoded by the HLA-DRB1 gene [34]. To date, 2690 distinct alleles of the HLA-DRB1, encoding 1899 proteins, have been identified (Figure 3).

The HLA-DRB1 gene consists of six exons, each encoding different protein domains (Figure 4). The Exon 2 of HLA-DRB1 is the most variable one and shares the amino acid sequence of the antigen recognition site.

Amino acid variations within different HLA-DR molecules were clustered into three major regions of hypervariability (HVR). The third region (HVR3) is encoded by Exon 2 and is located between amino acids 67–74 on the alpha helix of the HLA β1 chain, which form the most important site for primary T-cell recognition [35]. Allelic variations in HLA-DRB1 can result in HVR3 charge differences and can affect interactions with T cells. In RA, the impact of electric charge on disease susceptibility was shown. Amino acid motifs of the HVR3-carrying positive electric charge are associated with an increased risk of developing RA, whereas neutral or negative electric charge protects against RA [36,37].

Recently, a conditional haplotype analysis by Raychaudhuri et al. revealed that HVR1 region, formed by amino acids in positions 9–13, encoded by Exon 1, also strongly conferred ACPA-positive RA risk. Interestingly, amino acid residues 11 and 13 showed stronger association than any other polymorphic HLA-DRB1 amino acid position.

## 4. Shared Epitope Hypothesis

Over 30 years ago Gregersen et al. coined a hypothesis of a pathogenic role of three amino acid sequences (70QRRAA74, 70RRRAA74 or 70QKRAA74) located at positions 70–74, i.e., within the HVR3 of the DRβ1 chain, which form the so-called “shared epitope” (SE) [36]. It was hypothesized that the presence of these SE sequences allows the presentation of self-antigens to T lymphocytes, and thus plays a key role in the development of RA [36]. Recent reports confirm that the RA autoimmune process may be mostly triggered by the privileged binding of citrullinated peptides by HLA-DR molecules containing SE sequence [38].

The allele of the shared epitope (HLA-DRB1 SE) is present in 64–82% of patients with RA, which is significantly more than in their first-degree relatives (53.9–55%) and in the healthy control population (39–52%) [39,40,41]. The importance of a shared epitope in the development of the disease was confirmed in twin studies: RA developed in both twins 3.7 times more frequent when HLA-DRB1 SE was present and five times more frequent in pairs homozygous to SE as compared to pairs without SE [42]. The presence of the HLA-DRB1 shared epitope is very strongly associated with the development of ACPA-positive RA. HLA-DRB1 SE alleles show a significantly higher frequency in patients with anti-CCP (82–89.6%) than in anti-CCP-negative patients (53–70%) [40]. The presence of HLA-DRB1 SE alleles strongly affects the heritability of ACPA-positive RA, explaining 18% of the genetic variance in anti-CCP-positive RA, in contrast to 2.4% in ACPA-negative RA [10]. The presence of the SE allele is also associated with higher ACPA antibody levels [43]. Interestingly, a clear association between HLA-DRB1 SE and ACPA-positivity is that healthy individuals have not been found [44]. A recent study shows that this phenomenon can be at least partially explained by the impact of SE on ACPA-IgG V-domain glycosylation, not ACPA-positivity itself [30].

HLA-DRB1 SE has been also found to have relevant clinical importance by predisposing more destructive joint disease [45] and increased mortality [46]. Of particular interest, SE alleles were found more often among men, despite the fact that RA is more common in women [47]. The SE-coding alleles include:DRB1 *0401, *0409, *0413, *0416, *0421, *1419, *1421 (encoding 70QKRAA74 sequence),DRB1 *0101, *0102, *0105 *0404, *0405, *0408, *0410, *0419, *1402, *1406, *1409, *1413, *1417, *1420 (encoding 70QRRAA74)DRB1 *1001 (encoding 70RRRAA74) [48].

The greatest impact on the increase in the relative risk of RA is believed to have HLA-DRB1 *0404 (allele-odds ratio (OR) 3.5, 95% CI) [49]. The frequency of individual SE alleles varies depending on the age of onset of the first symptoms of the disease. Young-onset RA (≤40 years) is associated with the presence of DRB1 *0401 and *0404, while late-onset RA (≥60 years) is associated with DRB1*0101 [50]. The HLA regions in both young-onset RA and polyarticular juvenile idiopathic arthritis (JIA) patients show high similarity, and the common denominator is HLA-DRB1 *0401. This fact may indicate very similar pathomechanisms leading to the development of both diseases [51]. HLA-DRB1 *0405 is the most common SE allele in the Japanese population. Additionally, the relationship between the occurrence of HLA-DRB1 *0405 and serine in position 57 (non-SE residue) was shown. This amino acid residue also correlates with a higher risk of RA and is specific for the population of Asia [52].

The immunological role of SE probably is associated with an impact on adaptive immunity. Patients with SE are characterized by an increased expression of HLA-DR on B cells, which interact with T cell receptors. Consequently, it promotes CXCR4 expression on memory CD4+ T cells. The immunophenotyping analysis indicated higher frequency of memory CXCR4+CD4+ T cells in RA patients with at least one susceptible SE allele [53]. In addition, in the SE-positive patient, a significant increase in frequencies of both Th1 and Th17 lymphocyte subsets was observed [54].

## 5. HLA-DRB1 Alleles Other than the SE

It is estimated that over 20% of RA patients do not have the SE alleles [55]. The classic SE alleles determine only the shape of external binding site of the HLA-DR molecule. Therefore, shared epitope hypothesis does not explain the significance of allelic variations within HLA genes coding for the internal part of the binding groove. The identification of the new casual variants in HLA region is difficult due to its huge polymorphism. However, some protein variances outside the original SE motif were also identified as risk factors for RA. Raychaudhuri et al. analyzed the genome of 20,000 individuals, including over 5000 patients with anti-CCP positive RA, and identified gene variants contributing to RA risk, which encode two amino acids located in Positions 11 and 13 within HVR1 of the HLA-DRβ1 chain. While the role of Codon 13 in RA risk is still uncertain, Val 11 and Leu 11 have been reported to strongly influence the RA risk (OR = 3.8 and OR = 1.3, respectively) [56]. Moreover, a recent experimental model in macaques has shown that Val 11 and Phe 13 positions may be even more important than SE positions in the process of inducing the T-cell response against citrullinated peptides [57]. Val 11 has also been shown to correlate with the higher incidence of anti-CCP antibodies, similarly to SE variants [58].

The amino acids at Positions 11 (within HVR1), 71 and 74 (within HVR3) define 16 haplotypes. Val 11, Lys 71 and Ala 74 show the strongest association with the occurrence of RA. This amino acid sequence corresponds to the DRB1*0401 allele [56]. Some of the alleles overlap in terms of the amino acid sequences they encode (Table 1). The Lys 71 is associated with a high risk of RA and is present in patients carrying SE allele HLA-DRB1*0401. Nonetheless, this specific amino acid sequence occurs also in patients carrying other than SE-positive alleles, such as *1303 and *0301 [41].

Several studies indicate higher RA risk in individuals with the DRB1*0901 allele encoding a 70RRRAE74 motif in the HVR3 region [59]. HLA-DRB1*0901 allele frequency was significantly increased in RA patients without anti-CCP antibodies compared with controls and RA patients with anti-CCP antibodies [60]. The association with RA has been reported both in Caucasian and Korean populations [61].

## 6. Protective Alleles

The DERAA (D = aspartic acid, E = glutamic acid, R = arginine, A = alanine) sequence of amino acids at Position 70–74 in the HVR3 of the DRβ1 chain is considered the most important protective factor for seropositive RA, however, the mechanism of this allele-based protection is unknown [62]. Several DERAA-containing alleles have been identified: HLA-DRB1*0103, *1102, *1103, *1301, *1302, *1304 and *0402 [63]. The meta-analysis of four European trials underlined the protective impact of HLA-DRB1*13. It suggested that the DRB1*13 allele, rather than the whole DERAA, is associated with a protective effect. In addition, among DRB1*13, only the DRB1*1301 was identified as protective [64]. The protective effect of the DRB1*1302 allele was confirmed in the Japanese study [65]. The significance of the DRB1*0402 remains a matter of controversy. This allele was identified as having a protective effect over a dozen years ago, however, it has not been confirmed in subsequent studies [37,66,67]. The protective effect of the Ser 11 of the DRβ1, corresponding to the HLA-DRB1*03 allele, was also demonstrated in a few reports [56,68]. However, this remains controversial, because a possible role of the DRB1*03 allele in the development of anti-CCP-negative RA was also reported [69].

## 7. Ethnic Differences

The link between the presence of SE sequence and the increased risk of RA development exists regardless of ethnic or racial origin. However, there are ethnic differences in the incidence of specific SE alleles:DRB1 * 0401, * 0404, * 0408 are more frequent in the Caucasian population [70],DRB1 * 0405 in Asian population [71],DRB1 * 1402 in Native Americans [72],DRB1 * 0401, * 0404, * 0405 in Latin American ancestry population [73,74],DRB1 * 0101, * 0102 in Israeli Jews population [75],DRB1 * 1001, * 0102, * 0405 in African Americans. Interestingly, the overall prevalence of the SE-coding HLA-DRB1 alleles in African Americans is much lower (25%) than in European populations [76,77].

Substantial ethnical differences have been found in the prevalence of non-SE alleles, for example, in East Asians, a population-specific association with the DRB1*09:01 and DRB1*0405/*0901 heterozygotes has been identified [78,79]. Interestingly, the protective influence of individual alleles also varies depending on ethnicity. Among Asians, the alleles containing DERAA are not stipulated as having a protective significance, as opposed to the HLA-DRB1 * 0301, * 0403, * 0406, * 0701 and * 1405 alleles [80]. In populations from southern Mexico, HLA-DRB1*08 shows a definite protective effect [81].

## 8. Peptide Binding Affinity

The antigen-binding groove of the HLA-DR molecule consists of nine pockets that interact with the bound peptide. The most important are pockets P1, P4, P6, P7, and P9, binding the side chains of the Peptide Residues 1, 4, 6, 7, and 9, respectively [82]. The positive electric charge of the P4 pocket, which is strongly associated with the occurrence of SE alleles, significantly affects the increased ability to bind citrullinated peptides [37]. Citrulline in the P4 pocket predisposes the formation of a stabilizing hydrogen bond with the lysine-71β/arginine-71β. The analysis of crystal structures of HLA-DR-citrullinated epitope complexes has revealed that the mode of peptide binding is highly conserved in the P4 pocket, regardless of the SE variant (HLA-DRB1*04:01/*04:04/*04:05) [38].

The electronegative P4 pocket tends to accommodate arginine instead of citrulline and is associated with RA-protective HLA-DRB1 alleles. The important difference between protective HLA-DRB1*04:02 and *0401 is glutamate instead of lysine at Position 71β (E71K). This results in a lack of the hydrogen bond in the peptide backbone at Position P5, which is highly conserved in other HLA-DR molecules. In addition, aspartic acid at Position 70β enhances the formation of a salt bridge with P4-arginine [83,84].

Despite the conserved orientation of citrulline in the P4 pocket, a study by Ting et al. showed that the binding affinities vary depending on the type of peptide ligands, possibly due to differences in amino acid positions outside the P4 pocket. Indeed, HLA-DRB1*0404 and *0405 exhibit P1 and P9 pocket specificities. HLA-DRB1*0404 is characterized by a glycine-to-valine substitution in Position 86β (G86V) at the P1 pocket, which hinders the binding of hydrophobic residues. This results in low affinity for citrullinated peptides possessing a tyrosine instead of glycine in the P1 position. Similarly, HLA-DRB1*0405 shows a different selection of peptides with P9 residues, presumably due to an aspartic acid-to-serine substitution in Amino Acid Residue 57β (D57S) at the P9 pocket, which forms extensive hydrogen bonds with aspartate, making the HLA-DR–protein complex less stable [38].

Within the P6 pocket, there are Amino Acid Positions 11 and 13, which also likely play a role in antigen presentation. Positions 11 and 13 are also located in the peptide binding region of the HLA class II molecule, suggesting their likely role in antigen presentation. Position 11 appears to be the most variable residue in Pocket 6 of the β chain, determining the binding specificity of that pocket. It is thought that the possibility of peptide binding may be conditioned by interactions between molecules, as it has been shown that hydrophobic polar state of amino acid residues at Position 11 alter the binding of an antigen [85]. The side chains of amino acids in Positions 71 and 74 are spatially close to those in Positions 11 and 13, likely indicating interactions between the polymorphic HVR1 and HVR3 regions [56]. The importance of van der Waals contacts between His 13 and P4-Cit is particularly suspected. This is consistent with the observation of an occurrence of a histidine-to-serine polymorphism in Position 13 in protective HLA-DRB * 13: 01 [56,86].

In the HLA-DRB1 SE, the preferential accommodation of citrulline over arginine is not only found in Pocket 4, but also in other pockets (Table 2). Interestingly, the enhanced capacity to bind citrullinated peptides has been demonstrated also in case of some HLA-DQ2, HLA-DQ7 and HLA-DQ8 variants, but the significance of these observations for the ability of antigen presentation and possible influence on RA pathogenesis remains unknown. HLA-DRB1*0301 is an interesting example of an allele being associated with both a protective effect on RA and a positively charged P4 pocket, in which peptide binding affinity is low despite the expected preference of uncharged citrulline residues. The explanation for this phenomenon probably lies in the preferential accommodation of positively charged arginine residues in the P6 and P9 pockets [38]. In line with the above uncertainties, we need more studies assessing the influence of arginine-to-citrulline conversion in distinct pockets in molecules other than SE HLA-DRB1.

## 9. Genetic and Environmental Risk Factor’s Interactions

### 9.1. Genetic Interactions

The risk of developing RA is determined by both alleles of the HLA-DRB1 gene, coming from the mother and father. Balandraud et al. analyzed the associations of specific risk genotypes and anti-CCP antibody status and identified 30 “high-risk” genotypes, of which 10 contained a “double dose” of alleles predisposing RA. The highest risk genotypes were: HLA-DRB1*0401/* 10 (OR = 28.2), *0401/*09 (OR = 15.3). A total of 27 genotypes were considered “low risk” including, i.e., HLA-DRB1*12/*13, *07/*08, *11/*14, *03/*03, *08/*11 (OR = 0.2). Interestingly, also in this group was the HLA-DRB1 *01/*13 genotype, containing an RA-associated SE allele. This phenomenon was explained by the influence of the second protective allele. The contribution of the second allele is also visible on the example of genotypes with the HLA-DRB1*0401 allele-odds ratio depending on the concurrent allele ranges from 28.2 (HLA-DRB1*0401/*10) to 1.1 (HLA-DRB1*0401/*03) [67]. Anti-CCP positive RA may also be a result of the interaction between alleles of two different genes, as in the case of HLA-DRB1 SE and protein tyrosine phosphatase, a non-receptor type 22 (PTPN22) R620W A allele [40].

### 9.2. Smoking

In the RA pathogenesis, we can see an interaction between genes and environment, which may lead to trigger the disease (Figure 5). A strong interaction was found between HLA-DRB1 SE alleles and cigarette smoking, contributing to the development of anti-CCP positive RA [89]. Recent studies have shown that cigarette smoking affects not only positivity, but also higher levels of both ACPA and RF. Importantly, the impact of smoking on ACPA levels has been observed only in patients with SE alleles, while RF levels were elevated in all patients. The risk of future ACPA positivity and high levels have been attenuated by smoking cessation before the onset of RA [90]. It seems also likely that smoking in genetically predisposed individuals triggers immunity to citrullinated α-enolase [91]. Interestingly, additive interaction analyses showed that excessive salt intake among smokers more than doubled the risk of developing HLA-DRB1 SE-positive RA, indicating an important additive effect between these two RA risk factors [92].

### 9.3. Alcohol Consumption

Another recent observational study has indicated that alcohol consumption is dose dependently associated with a reduced risk of both ACPA-positive and ACPA-negative RA. Increased susceptibility of ACPA-positive RA has been shown to be a result of the interaction between risk factors, i.e., HLA-DRB1 SE and non-drinking [93].

### 9.4. Viral Infections

The Epstein–Barr virus (EBV) is another important environmental factor influencing the increased risk of RA development [94]. The analysis of the structure of the virus revealed the presence of the QKRAA sequence within the glycoprotein gp110. This sequence is also coded by the HLA-DRB1*0401 allele [95]. Moreover, the data from an in vitro study indicated that anti-CCP antibodies have an affinity to a deiminated protein encoded by EBV [96]. The cross-reactive autoimmune response resulting from molecular mimicry is a probable mechanism contributing to RA. Some studies have showed a synergistic effect between HLA-DRB1*04 and parvovirus B19 infection, but its plausible mechanism is unknown [97,98].

### 9.5. Periodontal Infections

In the context of RA, the role of infection by Porphyromonas gingivalis is also emphasized. This pathogen has been identified as a keystone in periodontitis, which is also associated with HLA-DRB1 alleles. P. gingivalis is unique among other periodontal pathogen in carrying peptidylarginine deiminase (PAD), an enzyme contributing to hypercitrullination of host proteins by deamination of C-terminal arginine. Similar cellular hypercitrullination induces pore-forming toxin leukotoxin-A (LtxA) produced by another periodontal pathogen, Aggregatibacter actinomycetemcomitans. Among HLA-DRB1 SE-positive individuals exposed on LtxA enhanced antibody response against citrullinated proteins was observed. This indicates a possible putative link between forming citrullinated neoantigens and driving autoimmunity in HLA-DRB1-SE—positive patients [99,100].

In animal models, HLA-DRB1 SE has an impact on the composition of the human gut microbiome. Recently, an association between HLA-DRB1 SE alleles and dysbiosis has been confirmed in humans but the mechanism is still unclear and there is a need for further studies [101].

## 10. Microchimerism and Non-Inherited Maternal Antigens

The vast majority of RA patients have at least one SE allele which may contribute to RA. However, some patients do not have any SE alleles. There is a number of reports supporting that the HLA-DRB1 gene affects the risk of developing RA not only as part of the patient’s genome, but also as an environmental factor. The exposure of pregnant woman to a fetal genotype probably contributes to the occurrence of RA in mothers, and this may partly explain the fact that RA is more frequent in women [102].

A small number of non-host stem cells can persist in another individual, which is called a phenomenon of microchimerism (Mc). The fetal and maternal cell exchange is common during pregnancy and may result in both the engraftment of fetal cells into the maternal organism (fetal Mc) and maternal cells into fetus (maternal Mc). Postnatal maternal Mc may also be the result of breastfeeding, as breast milk is rich in both cellular and soluble maternal HLA antigens [103]. Microchimeric cells persist in the host for many years [104]. In order to assess the incidence of Mc, the HLA-DRB1 gene was analyzed. The overall Mc frequency was 28.2%, fetal Mc was observed in 32.0% of the mothers, whereas maternal Mc in 23.4% of the newborns [105]. Some reports have implicated Mc onto the increased risk of developing autoimmune diseases, such as RA, type 1 diabetes, systemic lupus erythematosus, and systemic sclerosis [55,106,107,108,109]. Fetal Mc is probably one of the key phenomena affecting the development of RA in mothers without own SE alleles. In a study conducted by Rak et al. higher frequency and levels of HLA-DRB1*04, and HLA-DRB1*01 Mc were found in women with RA compared with healthy women. No differences were observed for other alleles, which are unrelated to RA development [102]. Cruz et al. also analyzed the impact of the child’s genotype on the mother’s risk of RA, by testing for alleles encoding SE and DERAA sequences, and Val 11, Lys 71, Ala 74 amino acids.

A three-fold higher risk was found in mothers with at least one child with SE allele, regardless of their own genotype. In addition, the risk was two-fold higher if the child had HLA-DRB1 risk alleles other than SE. Interestingly, having a child with the DERAA sequence was paradoxically associated with an increased risk of the disease in mothers. Another analysis conducted on a group of mothers without any of the risk alleles was partially consistent. It confirmed the fact of an increased risk of RA in women with a child with alleles coding for DERAA and Lys 71, but no statistically significant association was found in women with a child with alleles coding for the SE sequence, as well as Val 11 and Ala 74 [41]. The results associating RA to the exposure to DERAA alleles during pregnancy are consistent with the results of studies on the effect of DERAA sequence on T cell activation. Van Heemst et al. demonstrated that the epitope containing DERAA, derived from citrullinated vinculin proteins and from bacteria, is in vitro recognized by T cells. A similar activation of high affinity T cells probably occurs in pregnant women. Such a phenomenon was not observed in women with the HLA-DRB1*13 allele. The epitope with the core sequence DERAA encoded by HLA-DRB1*13 most likely induces an immunological self-tolerance in the mother, and thus protects against the development of RA [110].

Maternal Mc possibly affects the immune response of the child via the induction of fetal regulatory T-cells (T-regs) which suppress fetal T-cell reactivity against non-inherited maternal antigens (NIMA) [111]. The modulating impact on the immune response against NIMA has been described mainly in patients who underwent organ or bone marrow transplantation [112,113,114]. However, the association between NIMA and increased risk of autoimmune diseases development in the child, including rheumatoid arthritis and type 1 diabetes, has also been described [115,116,117]. Guthrie et al. showed the association between HLA-DRB1*04 encoding NIMA and the increased risk of RA in women with young-onset RA [55].

## 11. Significance of HLA-DRB1 Methylation Status

Epigenetic factors affect the function of a gene by regulating its expression. One of the most important epigenetic processes is DNA methylation, which prevents the binding of transcription factors to the gene promoter and, consequently, inhibits the transcription process. There is increasing evidence that the lower methylation (hypomethylation) of HLA-DRB1 gene promoter, leading to HLA-DRB1 overexpression, is associated with a higher risk of developing various autoimmune diseases. The loss of DNA methylation has been shown to be involved in psoriasis and multiple sclerosis pathogenesis [118,119]. Consistently, the rs9267649 protective variant was associated with increased DNA methylation and lower HLA-DRB1 expression in multiple sclerosis [119]. Furthermore, hypomethylation of HLA-DRB1 loci also induces HLA-DRB1 expression in systemic lupus erythematosus CD8+ T cells [120]. The HLA-DRB1 methylation status is likely to be relevant to RA, also. However, to date, there is a lack of information on its pathogenetic and clinical significance.

## 12. Associations with Clinical Presentations

### 12.1. Mortality Risk

Some amino acid haplotypes in HLA-DRB1 can be useful for the stratification of patients in terms of long-term outcomes, i.e., all-cause mortality, risk of radiographic damage, laboratory measures of disease activity, and response to treatment. RA disease severity and overall prognosis have been also shown to be gene-dose dependent, which means that patients with two copies of the susceptibility alleles are at risk of worse prognosis compared to single-copy carriers [121,122]. The multivariable analysis of 16 haplotypes (defined by Amino Acid Positions 11, 71, and 74) performed by Viatte et al. identified increased all-cause mortality for carriers of the VKA haplotype (Val 11, Lys 71 and Ala 74) and decreased for the SEA haplotype (Ser 11, Glu 71 and Ala 74). The strongest genetic predictor of mortality is thought to be Valine at Position 11 [123]. Surprisingly, these findings were not replicated in a recent study by Zhao et al., in which higher mortality was associated neither with the presence of VKA, Val 11, nor SE, but only the SKA haplotype (Ser 11, Lys 71 Ala 74), but this may be at least partially explained by the differences in patient populations [124]. The presence of HLA-DRB SE alleles, in particular, the HLA–DRB1*01/*04 genotype and homozygosity for the HLA–DRB1*0401 alleles, also contribute to a higher risk of premature death, largely from cardiovascular disease [46].

### 12.2. Risk of Radiographic Progression

The exact mechanism leading to radiographic damage in RA is unclear, however, the link between presence of anti-CCP antibodies, high levels of C-reactive protein and higher risk of radiographic progression is well established [125]. HLA-DRB1 haplotypes influence CRP levels, which is mainly mediated by anti-CCP status. The heritability of RA radiographic progression is moderate, with the estimated heritability rate of 45–58% [126]. In previous studies, both the HLA-DRB1 SE, Val 11 and Leu 11 amino acids conferred a risk of a more destructive course of disease, which denotes the tendency to develop erosions earlier and more rapidly, whereas DERAA motif was associated with a lower risk of radiological progression [45,58,124]. Furthermore, Val 11 showed the strong association with clinical (high swollen joint count) and laboratory (elevated sera inflammatory levels) RA activity measures, in contrast to serine at this position which was associated with lower inflammatory markers [127]. Nevertheless, in the recent meta-analysis of seven GWAS of 2775 cases (both seropositive and seronegative) testing for associations with radiological damage, the only single-nucleotide polymorphism attaining genome-wide significance regardless of ethnicity was rs112112734, which is in close proximity to HLA-DRB1 and with strong linkage disequilibrium with the shared-epitope. The association was less significant when the researchers adjusted for the presence of the rheumatoid factor. It is worthwhile to note, that for the purpose of this study, SE was represented by rs660895, tagging the commonest SE-encoding HLA-DRB1*0401 allele, which was also significantly associated with the trait and showed the same direction of effect in all but one GWAS [128].

### 12.3. Extra-Articular Manifestations

RA is a disease characterized not only by progressive joint damage, but also by potentially life-threatening extra-articular manifestations. Although the precise pathogenetic mechanism leading to them is unclear, a key role is widely assigned to HLA-DRB1 polymorphisms [129]. In a study by Turesson et al., the genotype’s influence on the development of extra-articular manifestations, such as interstitial lung disease, vasculitis, neuropathy, Felty’s syndrome (the triad of rheumatoid arthritis, splenomegaly and neutropenia), polyserositis (pleuritis, pericarditis), scleritis and glomerulonephritis, was analyzed. The presence of any of the individual HLA-DRB1 SE alleles was not significantly associated with extra-articular RA overall. However, a statistically significant association of the DRB1*0401 allele and the DRB1 *0401/0401 genotype with Felty’s syndrome was identified. “Double dose” HLA-DRB1*04 SE genotypes were found to modestly increase the risk of vasculitis (OR 2.44) and the overall risk of extra-articular RA (OR 1.79) [130]. Gorman et al. showed a strong relationship of vasculitis and the following three genotypes containing a “double dose” of SE alleles: DRB1*0401/0401, DRB1*0401/0404 and DRB1*0401/0101 [131]. Furthermore, in another study, in patients with the DRB1*04 allele (in particular *0404), a disfunction of endothelium-dependent vasodilatation was observed [132]. These data indicate that the determination of HLA-DRB1 status may possibly be a prognostic factor for the risk of cardiovascular events.

### 12.4. Pulmonary Fibrosis

Pulmonary fibrosis, which is a result of the rheumatoid arthritis-associated interstitial lung disease, alongside cardiovascular disease, is the most important extra-articular feature increasing mortality in RA [133]. In recent years, a rapid advance in RA treatment has been made, however, the benefits of new drugs were not demonstrated in RA-associated lung disease, suggesting that the development of pulmonary fibrosis is based on a different pathomechanism than the one responsible for joint inflammation. The HLA-DRB1*07 allele may play a particularly important role in the development of this extra-articular manifestation. In the study conducted in the United Kingdom, the HLA-DRB1*07 was found to be significantly more frequent in the group of patients with RA and secondary pulmonary fibrosis than in the group of patients with RA alone [134]. Another article has emphasized the role of the HLA-DRB1* 1502 allele among Asians, which may indicate a slightly different mechanism in this population [135].

### 12.5. Follicular Lymphoma

The identification of certain amino acid haplotypes in the HLA-DRB1 region may be useful in predicting the risk of follicular lymphoma (FL) development in RA patients. Patients with RA are at a three- to five-fold increased risk of disease-related non-Hodgkin’s lymphoma (NHL) [136]. FL is an indolent subtype of NHL with an overlapping genetic background with RA, strongly associated with HLA class II region variations. Baecklund et al. identified three of the most FL amino acid-associated haplotypes in HLA-DRB1: Ser11-Ser13-Asp28-Tyr30; Leu11-Phe13-Glu28-Cys30; and Pro11-Arg13-Asp28-Tyr30, corresponding with HLA-DRB1*03, *11, *13, *14; HLA-DRB1*01; and HLA-DRB1*15, *16 allelic groups, respectively. SE alleles, linked with the Leu11-Phe13-Glu28-Cys30 haplotype, also increased the risk of FL occurrence. Additionally, a significant smoking and shared epitope status interaction has been identified in FL patients [137].

## 13. HLA-DRB 1 and Response to Treatment

### 13.1. Conventional Synthetic Disease-Modifying Anti-Rheumatic Drugs and Cyclosporine A

Joint destruction severity in RA depends not only on the influence of genetic and immunological factors, but may be modulated using synthetic or biological disease-modifying antirheumatic drugs. Testing for specific HLA-DRB1 amino acid positions may possibly enable better treatment personalization and cost optimization in the future. The treatment of RA is based on conventional synthetic disease-modifying anti-rheumatic drugs (csDMARDs), such as methotrexate, sulfasalazine, leflunomide, and antimalarial drugs (chloroquine, hydroxychloroquine). Methotrexate is considered the first-line therapy for most patients with RA with an estimated ACR 50 (at least a 50% improvement) response rate around 41% [138]. In patients with a lack of methotrexate efficacy, several combination therapies of csDMARDs may be used. Shared epitope alleles are associated not only with high RA activity and less likely DMARD-free remission, but also as a predictor of an insufficient response to csDMARDs, predominantly methotrexate [139,140,141,142]. The HLA-DRB1*04, especially the *0405 allele, has proven to be of particular importance for this effect in both European and Japanese populations (Figure 6) [143,144]. Data also show that in SE-positive patients, triple-DMARD therapy (consisting of methotrexate, sulfasalazine and hydroxychloroquine) brings more benefit than if treated with methotrexate alone (94% and 32% responders, respectively; *p* < 0.01), with no such difference in SE-negative patients [15]. It has been also shown that aggressive immunosuppressive treatment in the SE-positive group is more effective if applied early [14]. Apart from SE, in a single study concerning the Pakistani population, HLA-DRB1*03 was found to be significantly associated with non-responders to methotrexate, but later, meta-analysis failed to confirm this observation [145,146].

Patients positive for HLA-DRB1*04 (especially with HLA-DRB1*0401/*0404 genotype) are also shown to be more likely to be treated with cyclosporine A (CsA), an immunomodulatory agent occasionally used in severe rheumatoid arthritis (Figure 6). This observation is consistent with the result of another study, in which CsA was reported as much more effective in the HLA-DRB1*04-positive as compared to *04-negative group (52.2% vs. 5.9%, respectively) [144,147].

### 13.2. TNF- α Blockers

Biological drugs are cornerstones of contemporary RA treatment strategy and TNF- α inhibitors (i.e., infliximab, adalimumab, etanercept, golimumab, certolizumab pegol) are the most commonly used. Around 68% of patients treated with anti-TNF-α agents and methotrexate achieve at least moderate response, but still, around one-third fail to respond [148]. The lack of efficacy can be divided into primary failure, assessed directly, usually 12 weeks after the start of treatment, and secondary failure, developing in initial responders during the course of therapy, which is commonly explained by the formation of anti-drug antibodies. To date, many research efforts have been directed towards elucidating the potential mechanisms leading to TNF-α resistance. Broadening the knowledge regarding this phenomenon may provide a better selection of patients to treat with anti-TNF-α drugs. With regard to the HLA-DRB1 gene, most studies indicate the relationship between the occurrence of alleles as risk factors for the destructive course of RA and better response to TNF- α drugs. In a study analyzing a primary response (assessed three to six months after treatment initiation) in sixteen HLA-DRB1 haplotypes defined by amino acids at Positions 11, 71, and 74, in both a infliximab-, etanercept-, or adalimumab-treated cohort, the VKA haplotype was found to be a predictive genetic biomarker for a better response [123]. Furthermore, a study by Criswell et al. showed that HLA-DRB1*0404 and *0101 alleles, both of which encode SE, are associated with favorable responses to etanercept at 12 months [149]. Later, this was also confirmed by Murdaca et al. [150] These findings are in line with results of the OPTIMA study, in which the HLA-DRB1 SE copy number was significantly associated with clinical efficacy in patients treated with adalimumab at week 26 [151]. An additional link between HLA-DRB1 and TNF-α responsiveness was provided by Liu et al. In subjects treated with adalimumab, the carriage of HLA-DRB1*03 allele conferred an increased risk of developing anti-drug antibodies, whereas the carriage of the HLA-DRB1*01 was found to be protective [152]. The studies on associations between HLA-DRB1 variations and response to treatment have been summarized in Table 3.

### 13.3. Abatacept

Biologic agents utilized in RA include not only TNF-α inhibitors, but also abatacept (a protein fusing the extracellular domain of human cytotoxic T-lymphocyte-associated antigen 4 [CTLA-4]). The immunophenotyping of lymphocyte populations elucidated that the increased expression of the chemokine receptor CXCR4 on memory CD4+ T cells significantly correlated with better response to CTLA4-Ig treatment. Moreover, a higher frequency of memory CXCR4+CD4+ T cells was connected to SE-positivity [53]. Consistent with these results, recent analysis of head-to-head data indicated that patients positive for SE alleles benefit more from abatacept than adalimumab. Briefly, after 24 weeks of abatacept exposure in SE-positive patients, the proportion of subjects who achieved ACR 20, 50 and 70 responses were significantly higher compared to the adalimumab group [153]. These results are in line with the particular effectiveness of abatacept in the SE-positive group demonstrated in a previous Japanese study (Figure 6) [154].

## 14. The Challenges Affecting the Implement HLA-DRB1 Genotyping in Clinical Practice

### 14.1. HLA Genotyping

The genotyping for several HLA alleles is already used in practice in the context of selecting treatment. Testing for the HLA-B*58:01 allele prior to initiation of the allopurinol, due to the confirmed association with the elevated risk for allopurinol hypersensitivity syndrome in Asian populations, has been taken into consideration in the 2012 American College of Rheumatology Guidelines for Management of Gout. Additionally, in the Clinical Pharmacogenetics Implementation Consortium (CPIC) Guidelines, pre-emptive genotyping for HLA-B*57:01 in the case of abacavir and HLAB*15:02 in Asians, in the case of carbamazepine, is recommended [155,156,157,158].

To date, the genotyping of HLA has been applied neither in the diagnostics nor in the selection of moment and type of RA treatment. This is linked to the multigenetic character of the disease, lack of knowledge about the probable genetic linkages between causal alleles from different loci, as well as the extensive polymorphism of the HLA region, in particular the HLA-DRB1 gene, implicating a number of rare pathogenic alleles with variable penetrance.

In the majority of to-date research concerning the role of HLA-DRB1 risk alleles, HLA typing at the amino-acid level (four-digit) was performed, with the use of polymerase chain reaction (PCR) sequence-specific oligonucleotide (SSO) probing or Sanger sequencing–based typing (SBT). Nevertheless, over the last few years, high-resolution HLA typing using next generation sequencing (NGS) technology and whole genome/exome sequencing (WGS/WES) data, is becoming more and more accessible, enabling highly accurate, allele-level HLA typing. The hyperpolimorphism and sequence similarity in the HLA region might hinder WGS-based HLA genotyping result interpretation. In order to mitigate this effect, NGS is accompanied with novel software tools, aligning sequence reads with alleles registered in the database, e.g., HLA-VBSeq, PHLAT, HLAminer, HLAscan enable the accurate causal inference of HLA genotypes [159,160].

### 14.2. Non-Mendelian Inheritance Pattern of RA–A Problem to Solve

In complex diseases like RA, the identification of rare functional variants with incomplete penetrance may be an essential issue to find common risk haplotypes.

The advent of NGS technology and WGS/EGS has provided new possibilities for the identification of new rare disease-causing genes and their variants, as well as determining whether the gene is inherited dominantly or recessively, in pedigrees with a disease following typical Mendelian inheritance. In the case of patients with highly penetrant forms of RA, WGS with variant calling enables the identification of causal alleles, even by the analysis of single individuals. However, the integration of WGS data with linkage analysis may further facilitate the mapping of genes responsible for RA in this group [161,162]. In RA, a rare, non-synonymous variant of the PLB1 gene has already been identified using this type of research [163]. Similarly, single rare risk variants were identified in other autoimmune diseases, e.g., systemic lupus erythematosus (SLE) and psoriasis [164,165].

The majority of patients with RA show complex and non-Mendelian inheritance patterns, even despite having familial clustering features. The identification of rare and low-penetrant causal variants in outbred populations is problematic and may require large patient collections to achieve sufficient statistical power [166]. However, in a cohort of unrelated individuals, WGS with identity-by-descent (IBD) mapping, a statistical method based on an analysis between pairs of unrelated individuals to measure the extent of haplotype sharing, may be useful to identify at-risk haplotypes. The selection of patients with haplotypes expected to contain causal variants highly improves the detection of variants with incomplete penetrance. Novel statistical methodologies, including non-parametric linkage analysis methods, enable a further assessment of rare variants in complex pedigrees, regardless of the mode of inheritance and estimated mutation penetrance [163].

### 14.3. Complex Pharmacogenetics of Anti-TNF Treatment Response

The phenomenon of resistance to the drug is both a crucial clinical problem which may result in the persistent high disease activity and the unnecessary risk of side-effects, as well as a pharmacoeconomic issue, exposing the health system to potential avoidable losses. A multitude of modern methods of RA treatment, which currently include drugs from the groups of anti-TNF-α and IL-6 blockers, B cell-depleting anti-CD20 antibody, as well as Janus kinase (JAK) inhibitors, foster the need of personalized therapy. The development of genotype-matched algorithms may be a key step in the further improvement of the risk–benefit ratio and cost-effectiveness of RA treatment. Anti-TNF-α inhibitors are the most commonly used biological drugs, hence the resistance to them has been the most widely analyzed.

The majority of existing analyses regarding the effectiveness of the anti-TNF-α treatment in RA have focused on the effects of a single SNP. Apart from the above-mentioned HLA-DRB1 variants, the influence of multiple genes’ polymorphisms on the effectiveness of applying anti-TNF-α has been described. They were within the following loci: TNF, TNFR1B (tumor necrosis factor receptor 2), interleukin-6 (IL-6), interleukin-10 (IL-10), TRAF1 (TNF Receptor Associated Factor 1), nuclear factor κB (NF-κB), encoding TLR signaling pathways (TLR2, TLR4, TLR5, CHUK, MyD88, IRAK3), Fc receptors for IgG immunoglobulins (FCGR2A, FCGR3A), NLRP3-inflammasome (NLRP3, CARD8), PTPRC (encoding protein tyrosine phosphatase), PDE3A–SLCO1C1 (encoding intracellular cyclic nucleotide signals regulator), CD84 (encoding B cell receptor), DHX32 (encoding putative RNA helicase), RGS12 (encoding regulator of G protein signaling), MICA (MHC class I polypeptide-related sequence A) [150,167,168,169,170,171,172,173,174]. Despite the fact that all anti TNF-α agents target the same cytokine, there are differences in the effectiveness of particular drugs in the various group of patients. Examples of these are the results of the several analyses of the role of TNF gene promotor polymorphism at Position –308, which showed that the –308GG genotype responded better to etanercept therapy than the –308AA genotype, which has not been observed among patients treated with infliximab. On the contrary, the better response to infliximab was associated with polymorphism at Position 238 [150,168,175,176]. However, in the extensive study including a total of 930 patients, such observations were not reproduced [122]. Interestingly, in a study by Padyukov et al., TNFA–308GG was not associated with better response to etanercept, however, the combination of TNFA–308GG with IL-10-1082AA genotypes showed better responsiveness [177].

In the light of above-mentioned facts, the observed associations between single SNPs and TNF-α blocker responsiveness should be interpreted with caution, as the candidate genes may form unclear clusters. Moreover, HLA-DRB1 susceptibility SNPs presumably constitute only a small part of the overall contribution for anti-TNF-responsiveness. Only an analysis of all the haplotypes, also covering candidate variants of HLA-DRB1, will enable us to use the knowledge about the role of various SNPs in clinical practice, including the choice of treatment with anti-TNF-α.

## 15. The Bumpy Road to Diagnostic Utility of HLA-DRB1

As we mentioned before, the occurrence of the HLA-DRB1 SE allele may trigger the formation of N-glycosylation sites in ACPA-IgG [30]. The presence of N-Linked Glycans in ACPA-IgG is already considered a promising biomarker of pre-clinical RA, as they could be detected up to 15 years before the first symptoms. Moreover, the intensity of ACPA-IgG N-glycosylation increases over time. The identification of patients with SE alleles could justify screening for N-Linked Glycans in this group to assess the risk of developing RA [30,31].

In a previous large Mendelian randomization study, which tested SNPs of various genes associated with IgG N-glycosylation, the only one associated with RA was rs9296009. In addition, modest association between rs9296009 and response to etanercept (measured by change in DAS28) was found. This SNP is present in the PRRT1 locus, located on short arm of chromosome 6 at positions 6p21.1–21.3, in linkage disequilibrium with rs660895 and rs6910071, both tagging HLA-DRB1*0401. The HLA-DRB1 variants were not included in the analysis [178]. In light of the above-mentioned facts, we conclude that HLA-DRB1 is the most likely to have a causal variant for extensive IgG N-glycosylation, which indicates the need for further studies. WGS-based HLA-DRB1 genotyping in a selected group of individuals with high ACPA-IgG variable domain (V-domain) glycosylation levels would be helpful.

The pinpointing of the HLA-DRB1 allele, which is responsible for pathogenicity of ACPA, may be a critical step to develop genetic test that can predict RA. It should be also noted that causal genes are likely to be clustered, therefore identifying the common risk chromosome 6 haplotype could be necessary. The potential genetic test could be useful, especially in an early phase of the disease as a valuable tool to complement and increase sensitivity of the currently used 2010 ACR/EULAR classification criteria, which score joint symptoms, serology (including RF and/or ACPA), symptom duration (whether <six weeks or >six weeks) and acute-phase reactants (CRP and/or ESR).

## 16. Concluding Remarks

Increases in the knowledge regarding the genetic variants within the HLA-DRB1 gene that significantly influence the risk of developing RA gives insight for a more comprehensive understanding of the interaction between B and T cells, which induce the T-cell response. In recent years, it has been clearly demonstrated that pathogenetic variations in HLA-DRB1 are not limited only to alleles encoding amino acid Positions 70–74, constituting SE, since Positions 11 and 13 are no less important, with this revelation aiding in the creation of new classification-ordering alleles. We know more and more about the complex interaction between the HLA-DRB1 SE and environmental risk factors, such as alcohol and smoking, and about the phenomenon of microchimerism, which can be the source of HLA-DRB1 risk variants among women with RA. Moreover, a number of variants of the HLA-DRB1 gene have been identified, which may shape response rates to individual drugs. Recent reports indicate that the preferential use of abatacept in patients with HLA-DRB1 SE may be an excellent example of such individualized therapy.

However, at present, the HLA-DRB1 allele typing is not widely used for both clinical and public health purposes. Nonetheless, the new era of NGS-based genome sequence analysis, accompanied by linkage analyses and evolving bioinformatic tools, opens the new, fertile avenue to identify rare, low-penetrant alleles, as well as common risk haplotypes. The HLA-DRB1 gene, which exerts the largest genetic contribution to RA in humans, will undoubtedly be crucial for the development of genotype-matched diagnostic and treatment protocols in RA patients.

## Figures and Tables

**Figure 1 cells-09-01127-f001:**
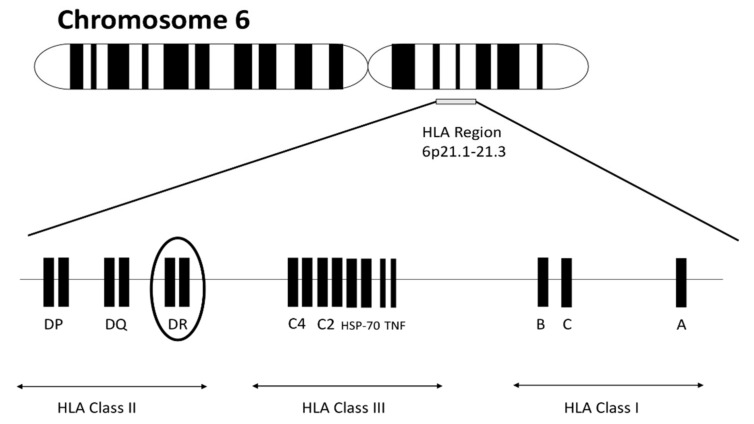
Human leukocyte antigen (HLA) molecules are encoded by three classes of genes located on short arm of chromosome 6 at positions 6p21.1–21.3. Within the DR subregion there is an outstandingly polymorphic HLA-DRB1 gene, which is of key importance in the pathogenesis of rheumatoid arthritis (RA).

**Figure 2 cells-09-01127-f002:**
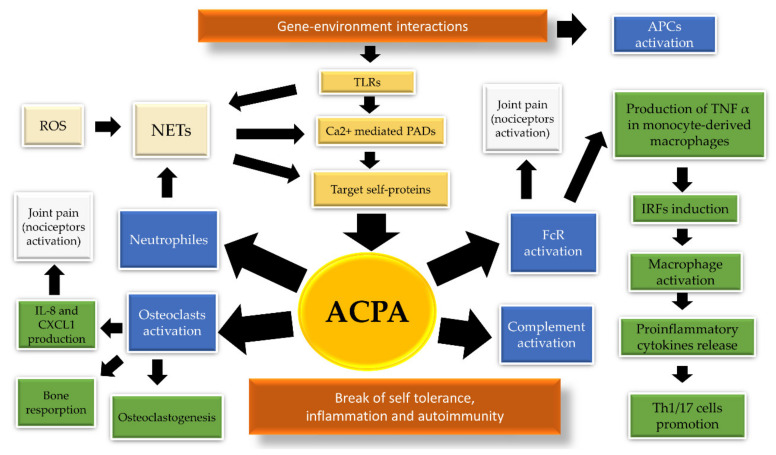
Various biological effects of anti-citrullinated protein antibodies (ACPA). The production of ACPA reflects break of immune tolerance and is dependent on the occurrence of both genetic, epigenetic and environmental factors. Genetic factors are: shared epitopes (SEs), protein tyrosine phosphatase non-receptor type 22 (PTPN22), α1-antitrypsin, type I interferons. Epigenetic modifications are: DNA methylation, histone acetylation and deacetylation, miRNA expression. Environmental factors include: noxious agents, influence of pathogens such as Porphyromonas gingivalis, Aggregatibacter actinomycetemcomitans (Aa) and Epstein–Barr virus (EBV). Interaction between genetic and environmental factors led to the activation of antigen presenting cells (APCs), such as dendritic cells, macrophages or B cells. Additionally, various noxious agents have a potential to activate toll-like receptors (TLRs). Triggering the innate immune response activate Ca^2+-^mediated peptidyl-arginine-deiminase (PAD) of granulocytes and macrophages, which catalyze citrullination of the target proteins located in the immune cells. The formation of neutrophil extracellular traps (NETs) may be induced by pathogens and reactive oxygen species (ROS). NETosis contributes to ACPA production by the externalization of citrullinated autoantigens and releasing of activated PAD, which form a pool of autoantigens that fuels autoimmunity. ACPA biologic effects rely on complement activation, osteoclasts stimulation, as well as direct macrophages and neutrophils activation. Joint pain may precede synovial inflammation and may be induced by ACPA via both osteoclast activation and direct Fc receptor binding [25].

**Figure 3 cells-09-01127-f003:**
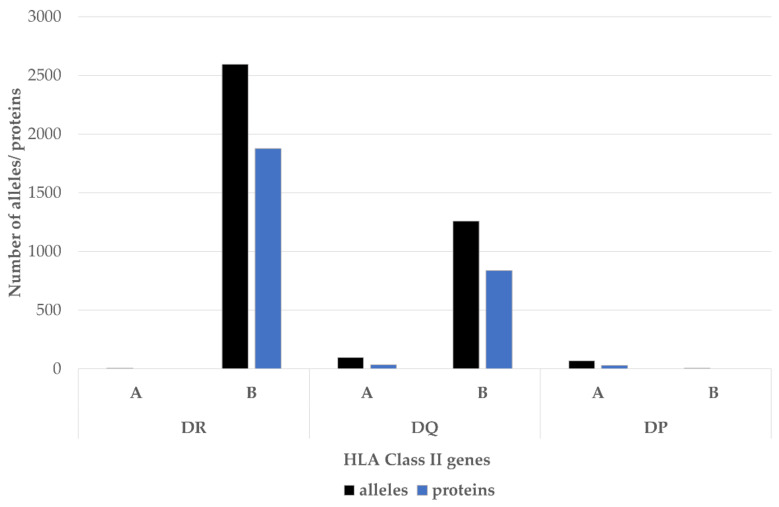
The HLA class II beta chains are much more polymorphic than the alpha chains. The most polymorphic HLA class II locus is HLA-DRB1-2690 alleles of this gene have been identified. Source: adapted from http://hla.alleles.org/nomenclature/stats.html; accessed on 21 April 2020.

**Figure 4 cells-09-01127-f004:**
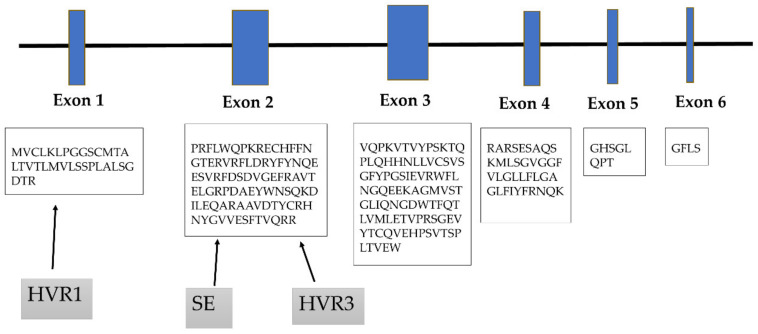
HLA-DRB1 gene is encoded by six exons. Exon 1 encodes the leader peptide, Exons 2 and 3 encode the two extracellular domains, Exon 4 encodes the transmembrane domain, and Exon 5 encodes the cytoplasmic tail. Amino acid motifs forming the first hypervariable region (HVR1) are encoded by Exon 1, the major HVR3, and shared epitope motifs are encoded by Exon 2.

**Figure 5 cells-09-01127-f005:**
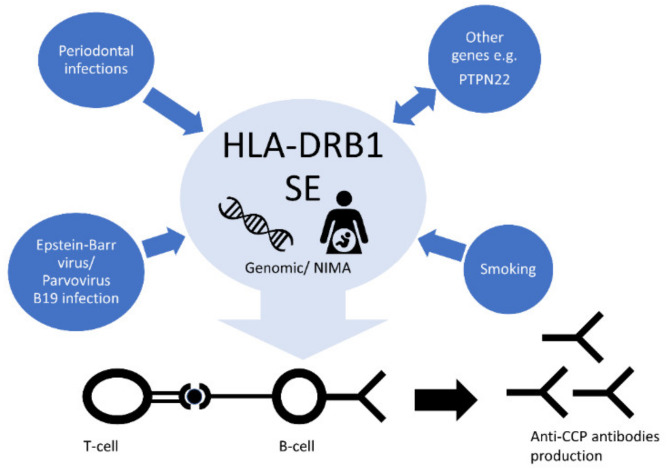
HLA-DRB1 interactions leading to the development of RA. By contributing to autoimmunity, the HLA-DRB1 SE alleles may interact with environmental factors such as smoking, Epstein–Barr virus (EBV), Porphyromonas gingivalis infection, as well as other susceptibility loci.

**Figure 6 cells-09-01127-f006:**
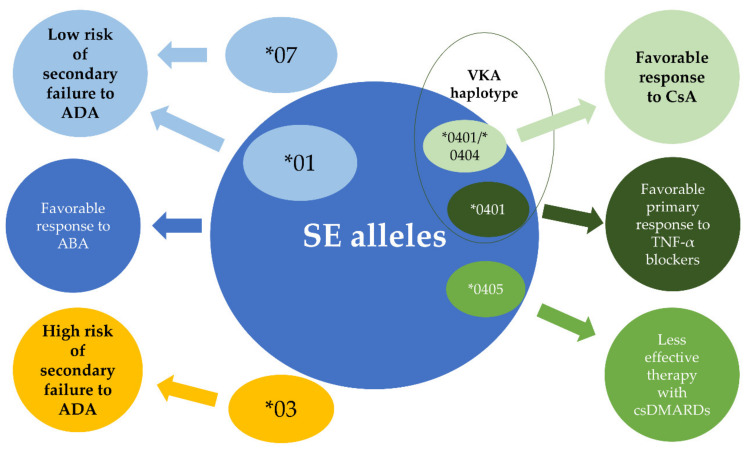
Potential HLA-DRB1 causal variants influencing specific treatment responses. Classical synthetic disease-modifying antirheumatic drugs (csDMARDs) include methotrexate, sulfasalazine, leflunomide, antimalarial drugs (chloroquine, hydroxychloroquine); CsA = cyclosporine; ADA = adalimumab; ABA = abatacept.

**Table 1 cells-09-01127-t001:** HLA-DRB1 alleles encoding amino acids at Positions 11, 71, 74 the “shared epitope” (SE) motif (associated with increased risk of RA) and the DERAA motif (associated with a reduced risk of RA). Several HLA alleles share overlapping amino acid sequences. The coexistence of the “shared epitope” motif and Val 11, Lys 71, Ala 74, is the most strongly associated with RA in Caucasians and it corresponds to the DRB1*0401 allele. Source: Data from Cruz GI, et al. Ann Rheum Dis 2017; 0: 1–6.

Allele	Ala 74	Val 11	Lys 71	SE	DERAA
*0101	+	-	-	+	-
*0102	+	-	-	+	-
*0103	+	-	-	-	+
*1102	+	-	-	-	+
*1301	+	-	-	-	+
*1302	+	-	-	-	+
*0401	+	+	+	+	-
*0404	-	+	-	+	-
*0405	-	+	-	+	-
*0408	-	+	-	+	-
*1001	-	+	-	+	-
*0402	+	+	-	-	+
*0403	-	+	-	-	-
*0407	-	+	-	-	-
*1101	+	-	-	-	-
*1103	+	-	-	-	-
*1501	+	-	-	-	-
*1502	+	-	-	-	-
*1104	+	-	-	-	-
*1201	+	-	-	-	-
*1601	+	-	-	-	-
*0301	-	-	+	-	-
*1303	+	-	+	-	-

**Table 2 cells-09-01127-t002:** Influence of selected HLA-DRB1 alleles and their molecular distinctions on binding affinity. HLA-DRB1 variants differ in case of P4 pocket net charge and preference for citrulline in other pockets of the binding groove. The occurrence of specific amino acid substitutions D57S at pocket P9 and G86V at pocket P1 results in slightly different hierarchies of binding in citrullinated self-peptides between SE allomorphs.

HLA-DRB1 Allele	Influence on RA Risk	P4 Pocket Charge	Preference for Cit over Arg Binding	Amino Acid Specificity	Binding Affinity of Selected RA Epitopes			Reference
					Vim-64,69,71Cit (59–71)	Vim-71Cit (66–78)	CII1240Cit	
*0401	risk allele	positive	P4; other unknown	no	low	high	high	[38,83]
*0402	protective	negative	Preference for Arg in P4; other unknown	E71K at P4 pocket	unknown	high	low	[83,87]
*0404	risk allele	positive	P4, P7	G86V at P1 pocket	low	high	low	[38,88]
*0405	risk allele	positive	P1, P4, P6, P9	D57S at P9 pocket	moderate	moderate	low	[38,88]
*0301	protective in Asians	positive	Preference for Arg in P6, P9	unknown	unknown	no binding	moderate	[87,88]

Peptides with IC50 values <1 μm, 1–5 μm, 5–250 μm, >250 μm were considered to have high, moderate, low and no binding, respectively. Cit = citrulline; Arg = arginine; Vim = vimentine; CII1240Cit = Collagen type II-1240Cit.

**Table 3 cells-09-01127-t003:** Studied concerning associations between HLA-DRB1 and treatment response.

Allele/Genotype	Treatment Response	f	Number of Patients (Male/Female)	Number of Patients Positive for Respective Variant	Number of Patients Anti-CCP-Positive at Diagnosis (%)	Additional Demographic Data	Reference
HLA-DRB1*0405	Inadequate response to csDMARDs	0.0003	124 (29/95)	64	85.5	Japanese population; mean disease duration 4.2 months; current/former smokers 19.3%	[143]
HLA-DRB1*0401/*0404	favorable response to CsA	0.016	54 (12/42)	4	unknown	Spanish population, Mean disease duration 12.1 years	[147]
HLA-DRB1*0401	favorable primary response to TNF-α inhibitors	0.007	1846 (432/1414)	1188	83	Data not shown	[123]
HLA-DRB1*03	high risk of secondary failure to ADA	0.006	634	37	unknown	Data not shown	[152]
HLA-DRB1*01	low risk of secondary failure to ADA	0.012	365	Data not shown	unknown	Data not shown	[152]
HLA-DRB1*07	low risk of secondary unresponsiveness to ADA	0.018	365	Data not shown	unknown	Data not shown	[152]
HLA-DRB1 SE	higher efficacy response with ABA vs ADA at week 24	Estimate of difference (95% CI) for DAS28 (CRP): 27.4	80	61	unknown	Mean disease duration 5.5 months	[153]
HLA-DRB1 SE	favorable response to ABA at week 24	<0.0001	72 (49/23)	47	89	Japanese population; mean disease duration 10.4 years	[154]

csDMARDs = classical synthetic disease-modifying antirheumatic drugs; CsA = cyclosporine; ADA = adalimumab; ABA = abatacept.

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
