# Peer review of "Current Understanding of an Emerging Role of HLA-DRB1 Gene in Rheumatoid Arthritis–From Research to Clinical Practice"

_cells, 2020, doi:10.3390/cells9051127_

Round 1
Reviewer 1 Report
This review summarizes the role of genetics, and specifically that of HLA-DR to the development of rheumatoid arthritis (RA).
- I wish the authors would elaborate further and even interpretate further the pathogenic potential of HLA linkage to RA.
- Furthermore, the potential of HLA linkage in RA in personalized medicine. Other than general comments I suggest to give more examples and discuss ways to study the potential of genetics in choosing the proper treatment.
- I disagree that treatment should apply to susceptible subjects in a preclinical stage. No one knows if those subjects ultimately are going to develop RA. Alternatively, I suggest to discuss the potential use of HLA screening within the recent ACR criteria for early RA.
- in section 12, page 11 - cyclosporine is not a DMARD based on the ACR criteria. Cyclosporine, like azathioprine and gold, are excluded from being DMARDs.
Minor comments:
- in page 2, first paragraph: there are 2 sentences that are too long and need rephrasing.
- that there are different HLA classes needs to be said and that HLA class II presents antigens to CD4 T cells.
- Replace "What is more" (for example Furthermore).
- in Section 7: Ethnic differences - unless this is the Journal's rules, add on the number of references instead of the name of first author and year.
- in figure 3 - the addition of 'alcohol consumption' is somewhat confusing
- cyclosporine is excluded as a DMARD based on the ACR criteria.
Author Response
Response to Reviewer 1 Comments
Point 1: This review summarizes the role of genetics, and specifically that of HLA-DR to the development of rheumatoid arthritis (RA).
Response 1: Thank you very much for the Reviewer’s kind comments.
Point 2: I wish the authors would elaborate further and even interpretate further the pathogenic potential of HLA linkage to RA.
Response 2: According to the suggestion we added section 3, in which we elaborate on further about a pathogenic link between HLA-DRB1 gene and ACPA. We describe a possible induction of IgG-ACPA N-glycosylation by HLA-DRB1 allele, which possibly lead to the pathogenicity of ACPA.
Point 3: Furthermore, the potential of HLA linkage in RA in personalized medicine. Other than general comments I suggest to give more examples and discuss ways to study the potential of genetics in choosing the proper treatment.
Response 3: As the Reviewer suggested, in section 14 we described pharmacogenomics issues. We have indicated the current use of HLA genotyping in diseases other than RA. Based on the example of anti-TNF-alpha drugs response studies, the advantages of testing the association of whole haplotypes instead of single SNPs / alleles, were indicated.
Point 4: I disagree that treatment should apply to susceptible subjects in a preclinical stage. No one knows if those subjects ultimately are going to develop RA. Alternatively, I suggest to discuss the potential use of HLA screening within the recent ACR criteria for early RA.
Response 4: Thank you for this remark. We agree that the prospect of applying the treatment in the preclinical phase of RA is controversial. This sentence has been deleted. In section 15 we disscuss the benefits (sensitivity increase) of attaching a genetic marker to the 2010 ACR / EULAR criteria, which only include the assessment of clinical and serological markers of the disease.
Point 5: In section 12, page 11 - cyclosporine is not a DMARD based on the ACR criteria. Cyclosporine, like azathioprine and gold, are excluded from being DMARDs.
Response 5: We agree with the Reviewer’s comment. We removed this information.
Point 6: In page 2, first paragraph: there are 2 sentences that are too long and need rephrasing.
Response 6: The indicated paragraph has been corrected.
Point 7: That there are different HLA classes needs to be said and that HLA class II presents antigens to CD4 T cells.
Response 7: Thank you for this remark. The introduction has been elaborated by adding indicated information regarding HLA.
Point 8: Replace "What is more" (for example Furthermore).
Response 8: The indicated phrase has been corrected.
Point 9: in Section 7: Ethnic differences - unless this is the Journal's rules, add on the number of references instead of the name of first author and year.
Response 9: The references have been corrected in section 7.
Point 10: in figure 3 - the addition of 'alcohol consumption' is somewhat confusing
Response 10: The figure 3 has been corrected by deleting ‘alcohol consumption’.
Point 11: cyclosporine is excluded as a DMARD based on the ACR criteria.
Response 11: Please see our response to point 5.
Reviewer 2 Report
The manuscript is interesting and well written. I suggest to briefly discuss the role of pharmacogenomics for choosing TNF Alpha inhibitors (see and ass references papers by Murdaca et a.:
- Murdaca G, et al . Pharmacogenetics and future therapeutic scenarios: what affects the prediction of response to treatment with etanercept? Drug Dev Res. 2014 Nov;75 Suppl 1:S7-S10.
2. Murdaca G, et al. Pharmacogenetics of etanercept: role of TNF-α gene polymorphisms in improving its efficacy. Expert Opin Drug Metab Toxicol. 2014 Dec;10(12):1703-10.
3.Murdaca et a. TNF-α Gene Polymorphisms: Association With Disease Susceptibility and Response to anti-TNF-α Treatment in Psoriatic Arthritis. J Invest Dermatol. 2014 Oct;134(10):2503-2509.
Author Response
Response to Reviewer 2 Comments
Point 1: The manuscript is interesting and well written. I suggest to briefly discuss the role of pharmacogenomics for choosing TNF Alpha inhibitors (see and ass references papers by Murdaca et al
Response 1: Thank you very much for the Reviewer’s kind comments. According to the Reviewer’s suggestion we added section 14.3 regarding the pharmacogenomics of TNF alpha blockers. We took into account indicated papers by Murdaca et al.
Reviewer 3 Report
This review is focused on the HLA-DRB1 and rheumatoid arthritis (RA)
This review is well written and easy to follow.
The significance of ACPA (anti-citrullinated protein antibody) and CCP (Cyclic Citrullinated Peptide) should be explained at their first appearance. A figure on the relevance of these antibodies in RA could further aid the reader to understand the main message of this manuscript. The authors should check whether any other abbreviation is explained.
Also, a figure depicting the treatment (especially for biological drugs) in the context of DRB1 patients can aid to understand the target of the therapy.
Finally, it is evident that HLADRB1 is relevant for RA but it seems that not all HLADRB1+ individuals will develop RA and several HLADRB1- individuals will develop RA. Are the additional factors possibly co-involved in the RA different from those associated with HLADRB1?
Author Response
Response to Reviewer 3 Comments
Point 1: This review is focused on the HLA-DRB1 and rheumatoid arthritis (RA). This review is well written and easy to follow.
Response 1: Thank you very much for the Reviewer’s kind comments.
Point 2: The significance of ACPA (anti-citrullinated protein antibody) and CCP (Cyclic Citrullinated Peptide) should be explained at their first appearance.
Response 2: According to the suggestion, we added section 3, in which we elaborate pathogenic importance of ACPA. We described the potential causal linkage between HLA and ACPA.
Point 3: A figure on the relevance of these antibodies in RA could further aid the reader to understand the main message of this manuscript.
Response 3: According to the suggestion, we added a figure 2 on page 3, which shows various biological effects of ACPA
Point 4: The authors should check whether any other abbreviation is explained.
Response 4:. As the Reviewer’s suggested, we verified whether all abbreviations are explained.
Point 5: Also, a figure depicting the treatment (especially for biological drugs) in the context of DRB1 patients can aid to understand the target of the therapy.
Response 5: As the Reviewer’s suggested, we added a figure 4 on page 16, depicting the identified influence of HLA-DRB1 alleles/genotype/haplotype on responsiveness to different drugs.
Point 6: Finally, it is evident that HLADRB1 is relevant for RA but it seems that not all HLADRB1+ individuals will develop RA and several HLADRB1- individuals will develop RA. Are the additional factors possibly co-involved in the RA different from those associated with HLADRB1?
Response 6: We deeply appreciate the Reviewer’s remarks. Of course, not all patients with risk HLA-DRB1 alleles will develop RA, because of the additional influence of environmental factors (e.g. smoking, viral and periodontal infections), as well as genetic factors (PTPN22), which interact with HLA-DRB1 and may be necessary to trigger the autoimmunity, as described in the manuscript. The contribution of second protective HLA-DRB1 allele may also be decisive.
It seems that influence of HLA-DRB1 risk alleles, in particular SE, is necessary to develop RA. However approximately 20% of RA patients are SE-negative. It can be partially explained by the impact of other than SE, HLA-DRB1 risk alleles. Moreover, some patients may have rare HLA-DRB1 risk allele which has not been identified yet. We cannot rule out the potential of epitope presenting by HLA-DQ (DQA1*03-DQB1*0301 or DQA1*03-DQB1*0302. Some polymorphisms associated with RA are also located in non-coding regions of the genome, but may possibly have a functional role, but further studies are needed. In women without any HLA-DRB1 risk variant the influence of fetal microchimerism (NIMA) should be taken into account, what is described in detail in the manuscript.
Reviewer 4 Report
The authors summarize current knowledge about the association of genetic HLA-DRB1 variants with rheumatoid arthritis.
Overall, the manuscript is well written and the review has a logic structure.
In the abstract and in the manuscript I am missing mechanistic aspects, hence for example if and how specific variants affect ligand-binding, structure, receptor abundance on membranes, antigen internalization, stability…. Most of the manuscript reports associations. Functional biological aspects are underrepresented.
It would be helpful to summarize the functional aspects in a table.
I am missing a short introduction of ACPAs, how /why citrullinated proteins are important for RA or other disease, and if/how ACPAs interfere with the target proteins' functions and composition of connective tissue matrix
A graphical presentation of the gene and protein of HLA-DRB1 with the localization of the most relevant variants would be helpful.
Abbreviations should be avoided in subtitles. Introductions to some abbreviations are missing, for example ACPA, NIMA, DERAA, anti-CCP
Paragraph 11. “Associations with clinical presentation” is very long and would profit from sub-structuring with subtitles
It is mentioned, that HLA-DRB1 is also associated with other diseases. How specific is HLA-DRB1 0404 for RA? Why do some patients develop RA and others vasculitis, MS, SLE etc.
Table 2: I suggest to add columns with number of patients, frequency of the respective variant carriers and frequency of ACPA positive versus negative patients, and info of demographic data (gender, age, smoking …) of the reported studies
The references need to be updated for studies published in 2020. There are no 2020 references
The conclusion jumps of methods of genotyping, which have not been discussed in the review. I suggest move this up and discuss thediagnostic value in the review, and shorten the conclusion.
Author Response
Response to Reviewer 4 Comments
Point 1: This review summarizes the role of genetics, and specifically that of HLA-DR to the development of rheumatoid arthritis (RA). The authors summarize current knowledge about the association of genetic HLA-DRB1 variants with rheumatoid arthritis. Overall, the manuscript is well written and the review has a logic structure.
Response 1: Thank you very much for the Reviewer’s kind comments.
Point 2: In the abstract and in the manuscript I am missing mechanistic aspects, hence for example if and how specific variants affect ligand-binding, structure, receptor abundance on membranes, antigen internalization, stability…. Most of the manuscript reports associations. Functional biological aspects are underrepresented.
Response 2: We deeply appreciate the Reviewer’s remarks. In order to fill the missing knowledge on the functional biological aspects of HLA-DRB1, section 8 has been added to the manuscript. We provided a detailed information about the influence of HLA-DRB1 alleles on electric charge of peptide-binding pockets, amino acid orientation, and binding affinities. The abstract has been also rewritten in order to incorporate these issues
Point 3: It would be helpful to summarize the functional aspects in a table.
Response 3: Table 2 has been added, summarizing the knowledge regarding influence of HLA-DRB1
Point 4: I am missing a short introduction of ACPAs, how /why citrullinated proteins are important for RA or other disease, and if/how ACPAs interfere with the target proteins' functions and composition of connective tissue matrix
Response 4: As the Reviewer suggested, we incorporated to manuscript section 2, in which biological effects of ACPA are summarized, and its possible role in RA pathogenesis is described.
Point 5: A graphical presentation of the gene and protein of HLA-DRB1 with the localization of the most relevant variants would be helpful.
Response 5: According to the Reviewer’s suggestion, a graphical presentation of HLA-DRB1 gene with the localization of the most hypervariable regions has been added
Point 6: Abbreviations should be avoided in subtitles. Introductions to some abbreviations are missing, for example ACPA, NIMA, DERAA, anti-CCP
Response 6: As the Reviewer’s suggested, we verified whether all abbreviations are explained.
Point 7: Paragraph 11. “Associations with clinical presentation” is very long and would profit from sub-structuring with subtitles
Response 7: According to the Reviewer’s suggestion, the indicated section (paragraph) has been divided into five, more clear subsections.
Point 8: It is mentioned, that HLA-DRB1 is also associated with other diseases. How specific is HLA-DRB1 0404 for RA? Why do some patients develop RA and others vasculitis, MS, SLE etc.
Response 8: HLA-DRB1*0404 allele occurs in high frequency (approximately 10%) in general population. Therefore it is more likely to evidence (achieve sufficient statistical power) the association between this allele and the disease compared to other, less frequent HLA-DRB1 risk variants. HLA-Apart from RA, DRB1*0404 allele is also considered to be associated with various autoimmune diseases, e.g. mixed connective tissue disease, diabetes, autoimmune hepatitis, giant cell arteritis. All these diseases are polygenetic and probable genetic linkages between causal alleles from different loci decide whether an individual will develop RA or vasculitis or MTCD.
Point 9: Table 2: I suggest to add columns with number of patients, frequency of the respective variant carriers and frequency of ACPA positive versus negative patients, and info of demographic data (gender, age, smoking …) of the reported studies
Response 9: As the Reviewer’s suggested, an additional columns: ‘number of patients (male/female)’, ‘number of patients positive for respective variant’, ‘number of patients anti-CCP-positive at diagnosis (%)’, ‘additional demographic data’, ‘p-value’, have been added to the table.
Point 10: The references need to be updated for studies published in 2020. There are no 2020 references
Response 10: According to Reviewer’s suggestions, the references have been updated with the papers published in 2020.
Point 11: The conclusion jumps of methods of genotyping, which have not been discussed in the review. I suggest move this up and discuss the diagnostic value in the review, and shorten the conclusion.
Response 11: The conclusion has been shortened. The diagnostic value of NGS-based high-resolution HLA typing has been discussed in section 14.
Round 2
Reviewer 1 Report
Congratulations!